Electrocortical theta activity may reflect sensory prediction errors during adaptation to a gradual gait perturbation

Jacobsen Noelle A. jacobsen.noelle@ufl.edu
Ferris Daniel Perry
J. Crayton Pruitt Family Department of Biomedical Engineering, University of Florida , Gainesville , FL , United States of America
Yakovenko Sergiy
Electronic publication date: 2024 Jun 5
Publication date: 2024
Volume: 12
Electronic Location ID: e17451
Received 2023 Nov 28; Accepted 2024 May 3
Copyright: ©2024 Jacobsen et al.
Copyright year: 2024
Copyright holder: Jacobsen et al.
License: This is an open access article distributed under the terms of the Creative Commons Attribution License, which permits unrestricted use, distribution, reproduction and adaptation in any medium and for any purpose provided that it is properly attributed. For attribution, the original author(s), title, publication source (PeerJ) and either DOI or URL of the article must be cited.
License URL: https://creativecommons.org/licenses/by/4.0/

Keywords: Human, Brain, Electroencephalography (EEG), Locomotion, Locomotor adaptation, Split-belt

Funding: The National Institutes of Health NIH R01-NS104772 NIH T32-NS082128 This research was supported by the National Institutes of Health (NIH R01-NS104772, NIH T32-NS082128). The funders had no role in study design, data collection and analysis, decision to publish, or preparation of the manuscript.

==============================
Locomotor adaptation to abrupt and gradual perturbations are likely driven by fundamentally different neural processes. The aim of this study was to quantify brain dynamics associated with gait adaptation to a gradually introduced gait perturbation, which typically results in smaller behavioral errors relative to an abrupt perturbation. Loss of balance during standing and walking elicits transient increases in midfrontal theta oscillations that have been shown to scale with perturbation intensity. We hypothesized there would be no significant change in anterior cingulate theta power (4–7 Hz) with respect to pre-adaptation when a gait perturbation is introduced gradually because the gradual perturbation acceleration and stepping kinematic errors are small relative to an abrupt perturbation. Using mobile electroencephalography (EEG), we measured gait-related spectral changes near the anterior cingulate, posterior cingulate, sensorimotor, and posterior parietal cortices as young, neurotypical adults (n = 30) adapted their gait to an incremental split-belt treadmill perturbation. Most cortical clusters we examined (>70%) did not exhibit changes in electrocortical activity between 2–50 Hz. However, we did observe gait-related theta synchronization near the left anterior cingulate cortex during strides with the largest errors, as measured by step length asymmetry. These results suggest gradual adaptation with small gait asymmetry and perturbation magnitude may not require significant cortical resources beyond normal treadmill walking. Nevertheless, the anterior cingulate may remain actively engaged in error monitoring, transmitting sensory prediction error information via theta oscillations.

Introduction

Human walking is highly adaptable, allowing humans to maintain stability with changing environments and conditions. The process is complex, depends on prior experience, and engages the brain and spinal cord (Ivanenko et al., 2013; Mirelman et al., 2018; Takakusaki, 2023). Yet remarkably, people easily and frequently adjust their gait every day to varying environmental and task conditions without giving it much thought. Damage to different brain regions can negatively affect this process and impair mobility (Malone & Bastian, 2014; Morton & Bastian, 2006; Vasudevan, Glass & Packel, 2014). Many therapeutic interventions aim to improve gait adaptation capability in occupational and rehabilitative settings, but restoring neurologically intact adaptation is often difficult. The specific neural mechanisms responsible for gait adaptation are not fully understood (Sato & Choi, 2022). To better understand how the nervous system adapts walking patterns, researchers have often used gait perturbations to study human walking across diverse conditions and tasks.

Research on human motor adaptation has often focused on two types of perturbation schedules: abrupt and gradual. Abrupt perturbations suddenly alter sensorimotor context or environments, causing large behavioral errors. One example of an abrupt perturbation is suddenly introducing a large visuomotor rotation during a computer mouse task such that pushing the mouse forward moves the cursor to the left (i.e., a 90-degree shift; Imamizu et al., 2000). During walking, researchers could introduce a large abrupt perturbation by applying a force to participants’ lower limbs during the swing phase with a cable that pulls the leg backwards (Kitatani et al., 2022). Conversely, gradual adaptation occurs when a perturbation is introduced slowly, resulting in smaller behavioral errors. In both examples described above for abrupt perturbations, there are ways to introduce gradual perturbations. During a computer mouse task, researchers could gradually introduce a visuomotor rotation by incrementally increasing the rotation angle by a few degrees after each movement attempt until it reaches 90 degrees. In the swing phase perturbation task, researchers could slowly increase the force on the cable pulling the swing leg backwards by a few Newtons after each step until it attains the desired large force magnitude.

Abrupt adaptation is assumed to be more effortful and explicit (i.e., volitional) than gradual adaptation due to the sudden and noticeable disturbance in the abrupt perturbation (Krakauer et al., 2019; Mazzoni & Krakauer, 2006). Adapting to gradual perturbations is often less difficult than abrupt perturbations but can still be effective (Bansal et al., 2023; Klassen, Tong & Flanagan, 2005; Sawers, Kelly & Hahn, 2013; Sawers & Hahn, 2013). Participants may not even be aware of a gradually introduced perturbation, so they must rely on more implicit (i.e., subconscious) processes to resolve sensory prediction errors (Krakauer et al., 2019; Mazzoni & Krakauer, 2006). These differences in awareness and error size could influence how the central nervous system attributes errors to the body or environment. They may also influence which brain areas are involved in the adaptation process.

Motor adaptation to abrupt and gradual perturbations is likely driven by fundamentally different neural processes. This is partly based on observations from upper- and lower-limb studies showing differences between abruptly and gradually introduced perturbations regarding the amount of adaptation (Torres-Oviedo & Bastian, 2012; Wong & Shelhamer, 2011), magnitude of after-effects (Kagerer, Contreras-Vidal & Stelmach, 1997), retention (Huang & Shadmehr, 2009; Klassen, Tong & Flanagan, 2005), and transfer (Kagerer, Contreras-Vidal & Stelmach, 1997; Kluzik et al., 2008; Malfait & Ostry, 2004; Torres-Oviedo & Bastian, 2012) of motor skills. During a force-field reaching task, abrupt perturbations were associated with changes in motor-evoked potentials, but gradual perturbations were not, suggesting that only abrupt perturbations induce changes in the primary motor cortex and corticospinal networks (Orban de Xivry et al., 2013). Additionally, disruption of the primary motor cortex through transcranial magnetic stimulation reduced early- and late-stage arm-reaching performance in the abrupt condition but had no effect on adaptation during the gradual condition (Orban de Xivry, Criscimagna-Hemminger & Shadmehr, 2011). In populations with central nervous system dysfunction, gait adaptation induced by small incremental perturbations led to better retention of the after-effect (Park et al., 2021; Tang et al., 2019) and transfer of treadmill to overground walking (Alcântara et al., 2018) than abrupt perturbations. This means that how perturbations are introduced to a patient is likely to influence how well they adapt and transfer motor skills, the latter being critical from a rehabilitation standpoint. Identifying potentially modifiable neural mechanisms that underpin motor adaptation is fundamental for developing effective interventions.

Measuring brain activity during motor adaptation tasks could increase our understanding of how the brain is adjusting to the new environment. For example, functional magnetic resonance imaging has revealed that a larger brain network appears to be involved in visuomotor adaptation to abrupt perturbations than gradual perturbations (Werner et al., 2014). Compared to upper-extremity motor adaptation, the neural correlates of human locomotor adaptation have been studied much less, primarily due to technical challenges associated with mobile brain imaging during locomotion. Studying the brain during locomotor adaptation is likely to provide insights into motor learning processes used to alter spatiotemporal elements of walking. Documenting changes in brain activity during human locomotor adaptation will aid in the development of rehabilitative gait therapies and assistive technologies (Gramann et al., 2014; Khajuria, Sharma & Joshi, 2022; Song & Nordin, 2021).

Advancements in high-density electroencephalography (EEG) now allow researchers to quantify human electrocortical activity during locomotion despite motion and muscle artifacts (for a recent review, see Song & Nordin, 2021). Because of its high temporal resolution, EEG can offer insights into neural correlates of behavioral effects observed during locomotor adaptation on a stride-by-stride basis. Accumulating evidence from mobile EEG studies suggest that electrocortical dynamics related to sensory integration and behavioral performance monitoring during locomotor adaptation may depend on the nature of the perturbation (Oliveira, Arguissain & Andersen, 2018; Peterson & Ferris, 2018; Peterson & Ferris, 2019; Solis-Escalante et al., 2020; Stokkermans et al., 2022) and the time course of adaptation (Jacobsen & Ferris, 2023; Wagner, Martínez-Cancino & Makeig, 2019). For example, sudden physical perturbations during standing and walking have been shown to evoke midfrontal theta synchronization (Peterson & Ferris, 2018) , or the analogous N1 potential, that scale with perturbation intensity (Aiden et al., 2019; Dietz, Quintern & Berger, 1984; Goel et al., 2018; Mochizuki et al., 2010; Solis-Escalante et al., 2020). Additionally, Solis-Escalante et al. (2020) demonstrated that neural oscillations near the midfrontal cortex scale with perturbation magnitude and predict reactive stepping behavior (Solis-Escalante et al., 2020). In our previous study, we found that an abrupt split-belt treadmill perturbation, where one belt moves faster than the other, elicited widespread changes in electrocortical activity that dissipated with practice (Jacobsen & Ferris, 2023). Early adaptation was associated with alpha (8–12 Hz) and beta (13–30 Hz) desynchronization in brain regions involved with sensory integration (posterior parietal) and motor execution (sensorimotor) and theta (4–7 Hz) synchronization in regions related to error-monitoring (anterior cingulate) and the default mode network (posterior cingulate). These changes during abrupt split-belt adaptation may be related, in part, to context-dependent adaptation, large kinematic errors, and heightened alertness. It is unclear how electrocortical activity evolves during gradual gait adaptation, when attentional demands and error size are attenuated.

The aim of this study was to better understand cortical mechanisms of error monitoring during gradual gait adaptation. Gradual gait adaptation is cognitively less demanding than abrupt adaptation (Sawers, Kelly & Hahn, 2013), possibly due to a reduction in error awareness or movement planning demands. We quantified electrocortical changes using mobile EEG as people adapt their gait to a gradual split-belt treadmill perturbation, which has been shown to produce smaller errors in stepping kinematics than an abrupt change in belt speed ratio (Torres-Oviedo & Bastian, 2012). We hypothesized that there would be no changes in gait-related theta power (4–7 Hz; with respect to normal tied belt, pre-gradual adaptation) in the anterior cingulate cortex, a region associated with error monitoring, because the kinematic errors and perturbation intensity would be small. This is based on evidence demonstrating that midfrontal electrocortical dynamics are associated with error awareness (Ficarella, Rochet & Burle, 2019) and change as a function of error size and practice (Anguera, Seidler & Gehring, 2009). In addition, we performed exploratory analyses in multiple cortical regions to investigate the effect of stages of gradual adaptation and error size on gait-related spectral differences.

Materials & Methods

Our primary aim was to determine how electrocortical activity changes across stages of gradual adaptation. Our secondary and tertiary aims were to explore how electrocortical activity is associated with kinematic error size (small versus large) and perturbation schedule (abrupt versus gradual).

Participants

Thirty-three young, neurotypical adults (15 female, 18 male) participated in this study. All participants self-reported being right-foot dominant (to avoid brain lateralization confounds) and physically active (regularly exercising ≥ twice per week for ≥30 min) and had normal or corrected to normal vision. None of the participants had a history of major musculoskeletal, neurological, or cardiovascular conditions. Participants were excluded if they had previously walked on a split-belt treadmill because the gait task needed to be novel. The experimental protocol was approved by the University of Florida Institutional Review Board (IRB201701603) and follows to the standards set by the Declaration of Helsinki. All participants provided written informed consent before participating.

Experimental design and procedure

Figure 1 shows a diagram of the experimental setup. Participants were equipped with our dual-electrode EEG system (described in the subsequent section), neck electrodes for electromyography (EMG), and motion capture markers. While participants walked on the split-belt treadmill, we recorded ground reaction forces from force plates under each belt (Fs = 1,000 Hz) and lower body kinematics using motion capture (Fs = 100 Hz; Optitrack, Corvallis, OR, USA). These data were synchronized through the motion capture software’s eSync. We used the Rizzoli lower body marker set, which was comprised of 26 markers, along with four rigid bodies placed on the shanks and thighs. Participants also wore earplugs to reduce their ability to use the sound of the treadmill to identify differences in belt speeds, which was particularly important for the gradual adaptation condition. They were also told to refrain from looking downward at the treadmill belts and keeping track of their cadence by any means (e.g., singing songs in head, counting) to prevent them from identifying subtle speed changes throughout the adaptation conditions. A fixation cross was placed on the board in front of them. Participants were told that the purpose of this cross was to help them stay oriented and prevent lateral shifting on the treadmill and that they did not have to stare at it during the entirety of the experiment.

Figure 1 Mobile brain and body imaging experimental setup.

Participants were equipped with a 128 dual-electrode EEG system covered with a black cap, eight neck EMG electrodes, and lower body motion capture markers. A force plate under each treadmill belt captured gait kinetics.

Our experiment had two adaptation conditions: abrupt and gradual. In both adaptation conditions, the final belt speed ratio was 2:1, with the belts moving at 1.2 m/s and 0.6 m/s under the right and left foot, respectively. We chose 1.2 m/s as our fast belt speed as it was close to the preferred treadmill walking speed of young adults (Terrier & Reynard, 2015) while likely keeping motion and muscle artifact contamination manageable. In the first adaptation condition, referred to as abrupt adaptation, we increased the error magnitude by abruptly perturbing the participants. During the second adaptation condition, referred to as gradual adaptation, we kept the error magnitude small by applying a gradual perturbation. We did not randomize the order of gradual and abrupt adaptation because this was a secondary analysis from a larger study that required the abrupt adaptation to be completely novel (Jacobsen & Ferris, 2023). However, results from Jacobsen & Ferris (2023) indicated that electrocortical (4–30 Hz) after-effects had washed out by the end of the first post-adaptation period.

Figure 2A shows the task paradigm. Participants walked on a split-belt treadmill (Bertec, Columbus, OH, USA) where each belt speed could be controlled independently. They were informed that the belt speeds would at some point be different, but not what the speeds would be. We told them when the treadmill was initially about to start or stop, but otherwise gave no indication of when speed changes would occur. We recorded all belt speed transition events with a button connected to the EEG system. To start, we recorded a pre-adaptation period in which participants walked with the belts moving together (i.e., tied) at three different speeds: slow (0.6 m/s), fast (1.2 m/s), and medium (0.9 m/s). During abrupt adaptation, participants encountered a sudden perturbation (a = 0.2 m/s2). The treadmill maintained a 2:1 belt speed ratio for the entire 15-min period. This was followed by a 10-minute post-adaptation period (post-adaptation #1) to wash-out after-effects, which is the minimum time suggested by Vasudevan, Hamzey & Kirk (2017). Roemmich & Bastian (2015) also used a 10-minute post-adaptation period to separate two different adaptation conditions (Roemmich & Bastian, 2015). Participants then experienced a gradual change in belt speeds. During the first five minutes of gradual adaptation, the right belt linearly increased the speed from 0.9 m/s to 1.2 m/s and the left belt linearly decreased the speed from 0.9 m/s to 0.6 m/s (a = 0.001 m/s2; similar acceleration to Long, Roemmich & Bastian, 2016). Once the belts reached terminal speed, the treadmill maintained a 2:1 ratio for two minutes. We limited the length of the gradual adaptation to keep entire experiment under 60 min; this was for the benefit of the volunteers and so that our results are comparable to clinical gait adaptation studies. Despite its brevity, we still expected motor adaptation to occur during gradual adaptation because after-effects can be present even after just two minutes of walking on a 2:1 split-belt treadmill (Roemmich & Bastian, 2015). The experiment ended with 5 min of tied-belt walking (referred to as post-adaptation #2) to wash out after-effects.

Figure 2 Experimental protocol diagram (A) and associated group-averaged step-length asymmetry results (B).

(A) The experiment includes two adaptation periods. During abrupt adaptation, the belt speeds abruptly split, and during gradual adaptation, the belt speeds incrementally split into a 2:1 ratio. (B) The gray shaded region indicates ±1 SD. The blue colored blocks roughly indicate the time window of each pre- and post-adaptation subcondition.

Dual-layer EEG and EMG setup

We used a custom dual-electrode system to conduct high-density EEG recordings. The system consisted of 128-pin type scalp electrodes and 128 flat-type “noise” electrodes using hardware from the BioSemi ActiveTwo system (Fs = 512 Hz; from Biosemi in Amsterdam, The Netherlands). For each scalp electrode, there was an inverted noise electrode physically attached (schematic in Fig. 3A). The noise electrodes were electrically isolated from the scalp electrodes, allowing movement artifacts and external noise to be exclusively recorded in separate channels. The design and functionality of the dual-electrode system was fully described in Nordin, Hairston & Ferris (2018); this previous work demonstrated a twofold enhancement in the signal-to-noise ratio through a phantom head simulation experiment.

Figure 3 EEG processing pipeline.

(A) This diagram of the dual-EEG setup shows how the scalp and noise electrodes were mechanically coupled. The inverted noise electrodes were covered with an additional artificial skin cap to mimic scalp conductivity. (B) Custom volume conduction models were created using the finite element method (FEM) with participants’ segmented magnetic resonance images (MRI). (C) Our EEG processing pipeline began with scalp EEG, noise, and EMG channel data. Preprocessing steps included motion and muscle artifact attenuation. After adaptive mixture component analysis, brain-like independent components were localized and clustered across participants. The dipole locations were normalized and plotted on the Montreal Neurological Institute template brain MRI. We then analyzed group electrocortical activity in the time-frequency domain to investigate how gait-related spectral power changes with stages of gradual gait adaptation. Note for color-blind readers: The centermost scalp topography circle indicates the region with greatest amplitude; amplitude diminishes radially outward according to the scalp topography lines.

We recorded neck EMG to isolate muscle activity and attenuate undesirable muscle signals in our EEG data. Richer et al. (2020) used a conductive phantom head and robotic motion platform to demonstrate the effectiveness of using neck EMG and dual-electrode EEG in improving recovery of complex artificial brain signals during motion (Richer et al., 2020). We used eight flat-type neck electrodes to record surface EMG signals from the left and right sternocleidomastoid, splenius capitis, and trapezius. The EMG electrodes had the same ground and reference as the scalp electrodes and were later rereferenced. We used a hard-wired 0.5 Hz square wave to synchronize the force plate, motion capture, EEG, noise, and EMG data. The data from the EMG and noise channels were used to attenuate artifacts in the EEG data, which we will elaborate on in the EEG Analysis section.

Multiple steps were taken prior to data recording to ensure robust EEG data were collected. Participants were instructed not to apply any hair products prior to the experiment. The placement of the EEG electrode cap followed the International 10–20 system. A reference line was marked on the forehead to monitor any potential shifting of the cap during the experiment. We recorded electrode locations using a 3D head scanner, with different scanners employed due to hardware and software issues (Eva scanner, Arctec3D, Santa Clara, CA, USA; Structure sensor on Apple® iPad; Scanner, Occipital Inc., San Francisco, CA, USA; itSeez3D). To establish good contact between the electrodes and the scalp, a blunt-tip gel syringe was used to gently part the hair and lightly abrade the scalp before applying the gel and placing the electrodes. The scalp electrode offsets were adjusted to be at or below 20 mV prior to recording. Once the scalp and noise electrodes were gelled, we positioned a secondary custom conductive cap (Eeonyx, Pinole, CA) over the noise electrodes to mimic the scalp electrodes’ circuitry and conductivity. This second cap was secured with stretch self-adherent tape to prevent shifting, and all wires were tightly bundled with Velcro straps to minimize cable sway, which is known to introduce artifacts into EEG signals (Symeonidou et al., 2018). Prior to recording, participants were shown the effects of their own jaw clenching, excessive eye blinking, and neck muscle tension on the raw EEG data stream and were instructed to refrain from these behaviors.

Behavioral analysis

To identify gait events, we used vertical ground reaction force data from the instrumented treadmill. The ground reaction force data were low-pass filtered (zero-phase, second order, Butterworth filter) with a cutoff frequency of 6 Hz to remove noise and down sampled to match the EEG sampling rate (512 Hz). We used a 10 N threshold to detect foot contact and foot lift-off events. We visually inspected time windows with abnormal gait event latencies (>4 standard deviations (SD)) and corrected any missteps (such as stepping on the wrong force plate).

We binned strides across the stages of pre- and post-adaptation to create subconditions shown in Fig. 2B. Pre-abrupt and pre-gradual adaptation were defined as the 30 strides preceding adaptation. The post-adaptation #2 period, which begins when the belts abruptly return to 1:1 ratio, was divided into initial post-adaptation (strides 1–10), early post-adaptation (strides 11–40), and late post-adaptation (last 30 strides).

Step length asymmetry was our primary metric of kinematic error because it adapts robustly during split-belt treadmill walking (Reisman, Block & Bastian, 2005). To quantify step length, we measured the distance between two ankle markers in the anterior-posterior direction at the instant the leading leg foot made contact with the treadmill belt. This led to two separate measures of step length, one for each leading limb. Fast step length refers to the step length when the limb on the faster moving belt was leading, and vice versa for slow step length. We quantified step length asymmetry as the difference in step length between the fast limb, which is on the faster belt, and the slow limb, which is on the slower belt, normalized by the sum of the two step lengths following Reisman, Block & Bastian (2005): (1) step length asymmetry=step lengthfast−step lengthslowstep lengthfast+step lengthslow.

We normalized step length asymmetry values by subtracting individual average step length asymmetry from the pre-abrupt adaptation condition (0.9 m/s tied-belt; i.e., within subject normalization) to compare behavior across participants.

We extracted asymmetric strides (i.e., kinematic errors) that occurred during adaptation so we could evaluate how electrocortical activity changes during stepping errors. We defined asymmetric strides as gait cycles with step length asymmetry values greater than two standard deviations from the mean of the last 30 strides of the reference condition preceding adaptation (i.e., pre-gradual adaptation): (2) SLAasymmtric>μreference+2σreference

where SLA is step length asymmetry and µ(σ) is the mean (standard deviation) of step length asymmetry from the reference condition. We used pre-gradual adaptation as the reference condition because in our previous study we found that behavioral after-effects during post-adaptation #1 did not fully washout (Jacobsen & Ferris, 2023). To determine the magnitude of the perturbation during the asymmetric strides, we calculated belt speeds using the average anterior-posterior velocity of the ankle markers during stance (when the foot was on the belt; Hoogkamer et al., 2015). We linearly interpolated gait variables and belt velocity missing values. Belt velocity data were smoothed using a finite impulse response filter (b = 1/6, a = 1). Belt velocity outliers were defined as elements more than three local standard deviations from the local mean within a six-element window. Outliers were then replaced with the mean of the strides preceding and following the outlier. We also calculated the belt symmetry in a similar way as step length asymmetry (following Hoogkamer et al., 2015): (3) belt symmetry=belt speedfast−belt speedslowbelt speedfast+belt speedslow.

Because participants had completed two adaptation periods (abrupt and gradual), we tested whether participants who were good at adapting to an abrupt perturbation were also good at adapting to a gradual perturbation. We used step length asymmetry to assess behavioral performance between abrupt and gradual adaptation. To quantify abrupt adaptation performance, we calculated the number of strides it took to reach steady-state step length asymmetry (similar to Fettrow et al., 2021; Finley et al., 2015). Steady-state was defined as the average value during the last 30 steps of abrupt adaptation. The threshold for reaching steady-state was defined as the point where step length asymmetry remained within 2 standard deviations of the steady-state for at least 10 consecutive steps. To quantify gradual adaptation performance, we used a separate performance metric because asymmetric strides were not occurring in an isolated period that was easily distinguishable from steady-state. We measured gradual adaptation performance as the percentage of asymmetric strides. Both performance metrics were normalized between zero and one and inversed so that a value of one indicated fewer strides to reach steady-state step symmetry or lower percentage of asymmetric strides for abrupt and gradual adaptation, respectively (i.e., a value of one indicates better adaptation). We performed linear regression to investigate the relationship between abrupt and gradual adaptation performance. The α-level was set at P = 0.05.

Volume conduction modeling

We constructed a volume conduction model of each participant’s head using individual structural magnetic resonance images (T1-MRI) following the Fieldtrip-SIMBIO pipeline (Vorwerk et al., 2018; Vorwerk et al., 2018; Vorwerk et al., 2018) outlined in Fig. 3B. Previous studies have demonstrated that realistic head models improve the accuracy of the EEG inverse solution (Akalin Acar & Makeig, 2013; Haueisen et al., 1997; Neil Cuffin, 1996; Roth et al., 1993; Vanrumste et al., 2002). We used the finite element method to construct the head model and merged it with digitized electrode locations using fiducial markers. The custom head models consisted of five compartment layers with specific tissue conductivity values: skin = 0.43 S/m, skull = 0.01 S/m, cerebral spinal fluid = 1.79 S/m, gray matter = 0.33 S/m, white matter = 0.14 S/m (Baumann et al., 1997; Haueisen et al., 1997; Hoekema et al., 2003; Vorwerk et al., 2019).

EEG Analysis

The EEG processing pipeline is outlined in Fig. 3C. We processed data in Matlab 2022a (MathWorks, Natick, MA, USA) using EEGLAB (v2022.0) (Delorme & Makeig, 2004), Fieldtrip (v20210614) (Oostenveld et al., 2011), and custom scripts as detailed in Jacobsen & Ferris (2023). We used a combination of standard and novel pre-processing steps to clean the EEG data (Fig. 3C, “channel-level analysis”). Data were high-pass filtered at 1 Hz (zero-phase, 2nd order, Butterworth) to attenuate slow drift. Scalp, muscle, and noise channels were separately re-referenced using the average across channels after temporarily removing bad channels (>3 SD). To attenuate undesirable motion and muscle artifacts, we used a data cleaning algorithm called iCanClean that uses strategically placed reference channels to remove noisy EEG subspaces (Downey & Ferris, 2023). iCanClean uses canonical correlation analysis to detect and reject components from the scalp channels that are highly correlated with components from the noise and muscle channels. Further details of how iCanClean was applied to this dataset were previously described in Jacobsen & Ferris (2023). We then used CleanLine (Mullen, 2012) to remove 60 Hz line noise. We removed bad channels based on statistical criteria: probability (criterion = 5), standard deviation (criterion = 500), and kurtosis (criterion = 5) (Gwin et al., 2011). We used clean_artifacts to reject noisy time windows and channels (from clean_rawdata v2.91) (Kothe, Miyakoshi & Delorme, 2019). On average, 119 channels (±6 SD) and 95% of frames remained in each dataset. The EEG data were re-referenced again to the common average and maintained full rank. EEG, ground reaction force, and motion capture data were merged into a single dataset using a hard-wired synchronization waveform (0.5 Hz).

Next, we transitioned to source-level analysis (Fig. 3C, “source-level analysis”). We used adaptive mixture independent component analysis to decompose the preprocessed EEG data, which amounted to an average of 50 minutes/person, into maximally independent components (Palmer, Z-Delgado & Makeig, 2012). We down-sampled the component data to 256 Hz and computed an equivalent dipole model for each independent component using the custom head models and Fieldtrip software (v20210614; Oostenveld et al., 2011). After source localization, we excluded artifact components based on the following criteria: positive power spectral density slope (linear slope [2–40 Hz] >0), high residual variance (>15%), localization outside the skull, low probability (<50%) of being a brain component per ICLabel (Pion-Tonachini, Kreutz-Delgado & Makeig, 2019), and high cross-frequency power-power coupling [quantified with PowPowCat; Thammasan & Miyakoshi, 2020] in specific frequency bands that are characteristic of eye and muscle components [correlation coefficient >0.3 in low (<8 Hz) and high (>30 Hz) frequency windows].

An average of 21 (±7 SD) components per participant were then used for group clustering using k-means. We estimated the optimal cluster number using evalclusters and the average of the results from the Calinski-Harabasz, silhouette, and Davies–Bouldin methods. We used two equally weighted spatial features for clustering, scalp topography and dipole location, and excluded time-frequency features to avoid potential inflation of the false-positive rate (Kriegeskorte et al., 2009). In cases where multiple components per participant were present within a cluster, we selected the components that explained the greatest amount of variance (i.e., the lowest independent component order number) to ensure that only one component per participant per cluster was considered, thus preventing artificial inflation of the sample size. Clusters lacking a dipole for a particular participant may reflect technical limitations of mobile EEG, such as contaminated independent components that did not meet our criteria to be classified as a “brain” component.

We performed time-frequency analyses for each dipole cluster. We computed event-related spectral perturbation plots and normalized them using single trial baseline removal (average log power across time within one gait cycle), which has been shown to be less sensitive to noisy trials (std_precomp parameters: cycles = [3,0.8], pad ratio = 2, basenorm = ‘on’, baseline = NaN (manual baseline subtraction applied later), trialbase = ‘full’) (Grandchamp, Delorme & Neri, 2011). In some analyses, we applied an additional baseline removal, which are described in the subsequent sections. We linearly time-warped event-related spectral perturbations to the group median gait cycle length using foot lift-off and contact events. We computed the average of all independent components’ event-related spectral perturbations within a cluster to create grand mean event-related spectral perturbation plots. We then calculated event-related spectral power for each subcondition to compare effects of error and perturbation magnitude. We examined spectral power changes in the frequency range of 2–50 Hz, which is slightly beyond theta, alpha, and beta bands, to gain a clearer picture of the extent of any spectral fluctuations and to visually inspect the data for signs of motion and muscle contamination.

Time course of gradual adaptation

To test our hypothesis that there would be no changes in theta power across gradual adaptation, we compared event-related spectral power (4–7 Hz) between pre-gradual adaptation and subconditions of gradual adaptation. Figure 4A shows how strides were binned during gradual adaptation conditions for a sample subject, with the associated belt speeds in Fig. 4B. The ramp-up period with a constant acceleration was broken down into three stages with 30 strides each: early ramp, mid ramp, and late ramp. We refer to the first 30 strides after the belts reached a full 2:1 speed ratio as the early 2:1 split condition and the last 30 strides as the late 2:1 split condition. We averaged event-related spectral perturbation data first at the participant level and then at the group level for each cluster of interest.

Figure 4 Sample participant stride binning from gradual adaptation period.

(A) Subconditions of gradual adaptation were created using 30-stride bins from the ramp-up period (early ramp, mid ramp, and late ramp) and the full 2:1 belt speed ratio period (early and late 2:1 split). (B) Approximate velocity of the left (solid line) and right (dashed line) belts during the gradual adaptation period.

We evaluated differences between pre-gradual adaptation and each subcondition (early ramp, mid ramp, late ramp, early 2:1 split, late 2:1 split) spectral power (2–50 Hz) using cluster-based permutation tests in Fieldtrip (Oostenveld et al., 2011) through EEGLAB (Delorme & Makeig, 2004), which uses the uses the Monte Carlo method to estimate the permutation p-value. Cluster-based permutation is a robust nonparametric statistic that both corrects for multiple comparisons and reduces potential for false negative effects (Maris & Oostenveld, 2007), making it a popular method for testing hypotheses in high-dimensional EEG data. For all cluster-based permutation tests, we used a significance threshold of 0.05 and 15,000 iterations. We used the maximum Cohen’s d from each significant pixel cluster as a measure of effect size (Meyer et al., 2021) and quantified the 95% confidence interval using bootstrapping (3,000 iterations). If there was an effect of condition on spectral power (2–50 Hz) in the anterior cingulate clusters, we then performed an additional cluster-based permutation test to evaluate the effect of condition (pre-gradual adaptation vs. gradual adaptation subcondition) on theta power (4–7 Hz) and test our hypothesis. As an exploratory analysis, we quantified spectral power changes (2–50 Hz) in the sensorimotor, posterior parietal, and posterior cingulate cortices and compared post-adaptation #2 subconditions (initial, early, late) with pre-gradual adaptation using cluster-based permutation tests.

Small versus large errors

We further categorized the asymmetric strides using a median split of step length asymmetry values, resulting in strides with large and small kinematic errors. On average 60 trials (range = [8,122]) of each error type went into each subject mean bin for the event-related spectral perturbation analysis. We calculated grand mean event-related spectral power across subjects during all large and small errors with respect to pre-gradual adaptation and then computed the difference in between large and small errors. We used cluster-based permutation tests to evaluate significant differences between each condition and pre-gradual adaptation and between both large and small error conditions.

Abrupt versus gradual adaptation errors

We tested if how a perturbation is introduced influences error-related electrocortical activity, when error and perturbation size are similar. We quantified differences in event-related spectral power between abrupt and gradual adaptation when the perturbation magnitude was equal and the behavioral error sizes were similar. This occurred when the belt speed ratios were the same (2:1), and step length asymmetry values were similar (within two standard deviations of the gradual adaptation asymmetric stride mean). We extracted the first 30 asymmetric strides during the full 2:1 split of gradual adaptation and matching asymmetric strides from abrupt adaptation. As an exploratory analysis, we used cluster-based permutation tests to evaluate the effect of condition (abrupt versus gradual) on event-related spectral power during these asymmetric strides. The caveat with this comparison is that the order of abrupt and gradual was not randomized because the primary focus of the study was to investigate brain dynamics during a novel abrupt split-belt perturbation (Jacobsen & Ferris, 2023). It is possible the previous practice of abrupt transition split-belt walking attenuated changes in electrocortical dynamics in addition to the effect of the gradual split-belt protocol.

Results

Participant demographics

Prior to any processing, we discarded datasets from three participants due to issues with EEG data recording (n = 2) or protocol errors (n = 1). The group analysis included data from the remaining 30 participants (n = 30), which included 15 females and 15 males, with a mean age of 22.8 ± 2.6 years (range = [19,29]) and a body mass index of 23.70 ± 3.85 (mean ± SD). Table 1 summarizes the participant demographics.

Step length asymmetry

We assessed the effect of split-belt adaptation on step length asymmetry. One participant displayed no change in step-length asymmetry during abrupt adaptation, as determined by the slope of the linear fit. Therefore, we classified them as an outlier and removed them from in all subsequent analyses. All remaining participants (n = 29) exhibited an increase in step length asymmetry during split-belt adaptation (Fig. 2B), with smaller asymmetry occurring during gradual than abrupt adaptation (p < 0.0001). Step length asymmetry became more symmetric over time in abrupt adaptation, and more variable over time in gradual adaptation. We observed after-effects following both abrupt and gradual adaptation, as indicated by the positive step length asymmetry during the early post-adaptation periods (Fig. 2B). Gradual adaptation notably lacks the rapid change in step length asymmetry seen in the initial abrupt adaptation, post-adaptation #1, and post-adaptation #2 periods (Fig. 2B).

Table 1 Participant demographics.

Sample Characteristics	n	%	Mean	SD	
Gender					
Female	15	50			
Male	15	50			
Age (years)			22.8	2.6	
Body mass index			23.70	3.85	
Notes.

N = 30

SD standard deviation

There was high inter- and intra-participant variability in timing of asymmetric strides during gradual adaptation. The average difference in treadmill belt speeds during the initial 30 asymmetric strides of gradual adaptation was 0.23 m/s, which means errors began to emerge before the belts reached a full 2:1 split. The belt symmetry during these initial errors was 0.55 ± 0.08 (mean ± SD).

Figure 5 shows the relationship between individuals’ normalized abrupt and gradual adaptation performance. A value of one indicates a good performance, meaning they were quick to reach steady state during abrupt adaptation or had a low percentage of asymmetric strides during gradual adaptation. The linear regression model revealed there was no correlation between abrupt adaptation performance and gradual adaptation performance (adjusted R2 = 0.01, p = 0.26). Therefore, the number of strides a participant took to reach steady-state step length asymmetry during abrupt adaptation was not predictive of the percentage of asymmetric strides they would have during gradual adaptation.

Figure 5 Abrupt and gradual adaptation behavioral performance.

Correlation between participants’ performance in terms of step-length asymmetry during abrupt and gradual adaptation. Abrupt adaptation performance was quantified using the number of strides to reach steady-state walking (y-axis). Gradual adaptation performance was evaluated using the percentage of asymmetric strides (x-axis). Performance metrics were normalized between [0,1] and inversed to compare between adaptation periods, with a value of one indicating fewer strides were needed to reach steady-state step symmetry or lower percentage of asymmetric strides occurred for abrupt and gradual adaptation, respectively. The solid black line indicates the linear regression fit, and the gray bars indicate the 95% confidence bounds.

Cortical clusters

The optimal k-means identified 12 cortical clusters, seven of which were included in this analysis because they were in brain regions of interest and included at least half of the participants. The location of cluster centroids within a standard brain image (Montreal Neurological Institute) are provided in Fig. 6. In Fig. 7, we provide more detailed information on each cluster, including the Talairach atlas label. Considering the inherent source localization error of EEG (∼1–2 cm according to Seeber et al. (2019); Sohrabpour et al. (2015)), we recognized that in certain cases, the Talairach atlas label may have overestimated the precision of the cluster centroid’s location within the brain. Therefore, we assigned new labels to some clusters (Fig. 7; “cluster label” column), which we will use throughout this paper. The posterior parietal cluster centroid was in the left hemisphere, but the individual sources were distributed between both hemispheres. We relabeled a cluster that the atlas called mid-cingulum as posterior cingulate. The cluster was very close to the posterior/midcingulate border, and a previous split-belt study that used a brain imaging technique with higher spatial resolution than EEG (positron emission tomography) found metabolic changes within the posterior, but not midcingulate cortex (Hinton et al., 2019).

Figure 6 Dipole cluster centroid locations.

We analyzed clusters localized to the sensorimotor, posterior parietal, and cingulate cortices. These clusters are plotted in axial (top left), sagittal (bottom left), and coronal (bottom right) planes on the Montreal Neurological Institute template brain image.

Figure 7 Locations of electrocortical clusters.

In the “Dipole locations” column, the large spheres are the cluster centroids, and the small spheres are the cortical dipoles plotted on the Montreal Neurological Institute template brain image. At least half of the participants are represented in each cluster. Only one dipole per participant was retained for analysis. Note for color-blind readers: The centermost scalp topography circle indicates the region with greatest amplitude; amplitude diminishes radially outward according to the scalp topography lines.

Time course of gradual adaptation

Figure 8A displays the belt speed ratio and magnitude of group-averaged step length asymmetry during subconditions of the gradual adaptation paradigm. These data are repeated for Fig. 9A so that the step length asymmetry (kinematic errors) and electrocortical dynamics can be visualized simultaneously for multiple brain regions. Mean step length asymmetry increased from pre-gradual adaptation to the early 2:1 split of gradual adaptation (Fig. 8A). During gradual adaptation, step length asymmetry was greatest during the late 2:1 split but was still about half the size of the after-effects during initial post-adaptation #2 (Fig. 8A).

Figure 8 Gait-related spectral power near the cingulate cortex across stages of gradual adaptation.

(A) The top row displays the belt speed ratio, and the bottom row shows the magnitude of group-averaged step length asymmetry. Step length asymmetry values were normalized by subtracting each participant’s pre-adaptation step length asymmetry. (B) Mean event-related spectral perturbations with respect to pre-gradual adaptation. Red indicates increased spectral power (neural synchronization) and blue indicates decreased spectral power (neural desynchronization), with respect to pre-gradual adaptation. Regions that are not significantly different from pre-gradual adaptation have a semi-transparent mask as determined by cluster-based permutation (α = 0.05). Vertical dashed lines indicate gait events: right foot contact (RFC), left foot-off (LFO), left foot contact (LFC), right foot-off (RFO). Circled regions of interest are discussed in the text.

Figure 9 Gait-related spectral power near the sensorimotor and posterior parietal cortices across stages of gradual adaptation.

(A) The top row displays the belt speed ratio, and the bottom row shows the magnitude of group-averaged step length asymmetry, repeated from Fig. 8A. Step length asymmetry values were normalized by subtracting each participant’s pre-adaptation step length asymmetry. (B) These plots display mean event-related spectral perturbations with respect to pre-gradual adaptation. Red indicates increased spectral power (neural synchronization) and blue indicates decreased spectral power (neural desynchronization), with respect to pre-gradual adaptation. Regions that are not significantly different from pre-gradual adaptation have a semi-transparent mask as determined by cluster-based permutation (α = 0.05). Vertical dashed lines indicate gait events: right foot contact (RFC), left foot-off (LFO), left foot contact (LFC), right foot-off (RFO). Circled regions of interest are discussed in the text.

Most cortical clusters (71%) did not exhibit changes in spectral power across gradual adaptation (Figs. 8–9). Figure 8B shows grand mean event-related spectral perturbations across stages of gradual adaptation with respect to pre-gradual adaptation; red indicates increased spectral power (neural synchronization) and blue indicates decreased spectral power (neural desynchronization). All regions without significant differences from pre-gradual adaptation, as determined by cluster-based permutation tests, have a semi-transparent mask. Cluster-based permutation tests revealed that the left anterior cingulate exhibited a significant difference in gait-related spectral power (2-50 Hz) from pre-gradual adaptation during the middle of the ramp-up (mid-ramp) of gradual adaptation (p = 0.010), before the belts reached a full 2:1 speed ratio. The effect size, as measured by the maximum Cohen’s d, was d = 1.59 (95% confidence interval (CI) = [1.21,1.95]). Descriptively, this positive pixel cluster (region 1 in Fig. 8B) was located near the theta and alpha bands. There were no significant differences in event-related spectral power when step length asymmetry magnitude was greatest, during the end of the split-belt period (late 2:1 split). Among the sensorimotor and posterior parietal clusters, only the posterior parietal cortex exhibited an effect of gradual adaptation subcondition on event-related spectral power from the cluster-based permutation tests. We found a positive pixel cluster during the late 2:1 split subcondition (region 3 in Fig. 9B; p = 0.019) whose effect size was d = 1.48 (95% CI = [1.06,1.95]).

Figure 10 shows the average theta power (4–7 Hz) near the left anterior cingulate across the gait-cycle for pre-gradual adaptation and mid ramp. A cluster-based permutation test revealed that there was an effect of condition on theta power (p = 0.033, d = 0.86, 95% CI = [0.55, 1.16]). There was greater theta power during the mid ramp condition than during pre-gradual adaptation.

Figure 10 Mean gait-related theta power (4–7 Hz) near the left anterior cingulate cortex.

Shaded regions indicate ± 1 SD. Dashed gray horizontal line indicates zero. Black bar and asterisk (*) indicates a significant difference across conditions based on cluster-based permutation (p < 0.05).

Visual inspection of the event-related spectral perturbations indicated that theta synchronization near the left anterior cingulate and alpha synchronization near the posterior parietal cortex increased over the course of gradual adaptation, from the early ramp to the late 2:1 split period (Figs. 8B and 9B, respectively). Theta synchronization near the left anterior cingulate cortex appeared to be strongest preceding left foot contact, during right leg single support. By the end of gradual adaptation, alpha synchronization near the posterior parietal cortex was occurring during the swing phase of the right leg in the posterior parietal cortex.

There were four cortical clusters that exhibited an effect of a post-adaptation #2 condition on event-related spectral power as determined by cluster-based permutation tests: the posterior cingulate, bilateral sensorimotor, and posterior parietal cortices. The right posterior cingulate exhibited desynchronization during the initial post-adapt. #2 (Fig. 8B, region 2; p = 0.003, d = 1.48, 95% CI = [1.03, 1.96]) and early post-adapt. #2 conditions (Fig. 8B, region 3; p = 0.023, d = 1.36, 95% CI = [0.86, 2.02]) with respect to pre-gradual adaptation. By late post-adaptation #2, the right posterior cingulate displayed synchronization (Fig. 8B, region 3; p = 0.004, d = 1.28, 95% CI = [0.87,1.90]). The abrupt transition back to tied-belt walking (initial post-adaptation #2) also evoked desynchronization with respect to pre-gradual adaptation near the left sensorimotor (Fig. 9B, region 1; p = 0.0004, d = 1.63, 95% CI = [1.18, 2.14]), right sensorimotor (Fig. 9B, region 2; p = 0.009, d = 1.40, 95% CI = [1.05, 1.82]), and posterior parietal cortices (Fig. 9B, region 4; p = 0.0028, d = 1.35, 95% CI = [0.98,1.85]) that dissipated by early post-adaptation #2. Descriptively, these enhanced desynchronizations were within the alpha and theta bands for the left sensorimotor cortex, the beta band for the right sensorimotor cortex, and the alpha and beta bands for the posterior parietal cortex.

Small versus large errors

We categorized the asymmetric strides as being a large or small error based on step length asymmetry. The magnitude of group-averaged step length asymmetry was 0.07 ± 0.02 for large errors strides and 0.04 ± 0.02 (mean ± SD) for small errors (Fig. 11A). Figure 11B displays the event-related spectral power during strides with large and small errors and the difference between the two conditions (large minus small) with respect to pre-gradual adaptation. Regions without significant differences between large and small asymmetric strides (Fig. 11B, third column) have a semi-transparent mask. There was only one significant effect found of condition on event-related spectral power, and this was near the left anterior cingulate source for strides with large asymmetry (i.e., large errors) with respect to pre-gradual adaptation (p = 0.015, d = 0.93, 95% CI [0.64, 1.70]). The location of this significant positive pixel cluster was approximately in the theta and alpha bands preceding left foot-contact (region 1 in Fig. 11B). There were no significant differences between small and large errors in any cortical cluster.

Figure 11 Gait-related spectral power during large vs. small errors of gradual adaptation.

(A) The violin plot displays the group step-length asymmetry magnitude mean (dot) and standard deviation (error bars). Step length asymmetry values were normalized by subtracting each participant’s pre-adaptation step length asymmetry. (B) These plots show changes in spectral power during large errors/asymmetry, small errors/asymmetry, and the difference between the two conditions (large minus small) with respect to pre-gradual adaptation. Please note that “large” and “small” are relative terms used to describe the top and bottom halves of the median split for kinematic errors in the gradual adaptation condition. All regions without significant differences from pre-gradual adaptation (first two columns) and between large and small errors (third column) have a semi-transparent mask. Vertical dashed lines indicate gait events: right foot contact (RFC), left foot-off (LFO), left foot contact (LFC), right foot-off (RFO). Circled regions of interest are discussed in the text.

Abrupt versus gradual adaptation errors

As an exploratory analysis, we compared asymmetric strides between abrupt and gradual adaptation when the perturbation and error magnitude were approximately equal. The group-average step length asymmetry magnitude during these asymmetric strides was 0.06 ± 0.02 for abrupt adaptation and 0.07 ± 0.02 for gradual adaptation (Fig. 12A). Figure 12B shows grand mean event-related spectral perturbations from abrupt adaptation errors (with respect to pre-abrupt adaptation), gradual adaptation errors (with respect to pre-gradual adaptation), and the difference between the two (abrupt minus gradual). Cluster-based permutation tests revealed enhanced synchronization during abrupt adaptation errors with respect to pre-abrupt adaptation in the left anterior cingulate (Fig. 12B, region 1; p = 0.011, d = 1.36, 95% CI = [0.99, 1.80]) and left posterior cingulate cortices (Fig. 12B, region 2; p = 0.014, d = 1.17, 95% CI = [0.94, 1.47]). For gradual adaptation errors, only the right posterior cingulate showed changes in spectral power, which decreased with respect to pre-gradual adaptation (Fig. 12B, region 4; p = 0.024, d = 1.53, 95% CI = [1.01, 2.20]). Cluster-based permutation tests indicated that there was an effect of subcondition (abrupt versus gradual) on event-related spectral power near the left posterior cingulate (p = 0.005, d = 1.41, 95% CI [1.02, 1.91]), right posterior cingulate (p = 0.004, d = 1.45, CI = [0.92, 2.13]), and posterior parietal cortices (p = 0.017, d = 1.47, 95% CI [1.06, 1.96]). The left posterior cingulate cortex (Fig. 12B, region 3) and right posterior cingulate cortex (Fig. 12B, region 5) showed enhanced synchronization during abrupt adaptation errors compared to gradual adaptation errors, whereas the posterior parietal cortex showed stronger desynchronization (Fig. 12B, region 6).

Figure 12 Gait-related spectral power during similar sized errors of abrupt and gradual adaptation.

(A) The top row shows the belt speed ratio, which indicates an equal perturbation magnitude. The bottom row displays the group step-length asymmetry magnitude mean (dot) and standard deviation (error bars), which indicates error size is approximately equal. Step length asymmetry values were normalized by subtracting each participant’s pre-adaptation step length asymmetry. (B) These plots show changes in spectral power during gradual adaptation errors (with respect to pre-gradual adaptation), matching abrupt adaptation errors (with respect to pre-abrupt adaptation), and the difference between the two conditions (abrupt minus gradual). All regions without significant differences from their respective baseline (first two columns) and between abrupt and gradual errors (third column) have a semi-transparent mask. Vertical dashed lines indicate gait events: right foot contact (RFC), left foot-off (LFO), left foot contact (LFC), right foot-off (RFO). Circled regions of interest are discussed in the text.

Visual inspection revealed more alpha and beta desynchronization near the posterior parietal cortex (region 6) and more theta synchronization near the bilateral posterior cingulate cortex (regions 3 and 5) during the abrupt asymmetric strides compared to the corresponding gradual asymmetric strides (Fig. 12B). In the left posterior cingulate, there was a positive (red) pixel cluster approximately within the theta band (region 3 circled in Fig. 12B) that occurred following left foot contact in the plot showing the difference between the abrupt and gradual errors; this was the limb on the slower belt. Similarly, there was a positive (red) pixel cluster observed in the right posterior cingulate during left leg single support (Fig. 12B, region 5). While there are differences between abrupt and gradual adaptation errors in both posterior cingulate clusters, the difference in the left posterior cingulate was caused by increased theta power during abrupt adaptation, whereas the difference in the right posterior cingulate was caused by decreased theta power during gradual adaptation. Lastly, alpha desynchronization near the posterior parietal cortex appears to occur throughout the gait cycle but is more prominent during right leg stance (Fig. 12B, region 6).

Discussion

Our results demonstrate that for young, neurotypical adults, gradual adaptation to a slowly changing belt speed ratio when walking on a split-belt treadmill elicits changes in neural oscillations even though participants never experience large stepping errors (as measured by step length asymmetry). While spectral power in most cortical clusters during the gait cycle remained unchanged, the left anterior cingulate did display gait-related theta synchronization during middle of the ramp-up period of gradual adaptation. Theta power near the anterior cingulate spectral was also greater than pre-gradual adaptation during strides with larger step length asymmetry (i.e., during relatively larger errors). Theta dynamics do not appear to proportionally change with error size for the low range of errors sizes experienced during gradual adaptation, as indicated by the lack of difference in spectral power between large and small errors. Together, these results suggest that although gradual split-belt adaptation does not appear to recruit many cortical resources beyond normal treadmill walking, the anterior cingulate still appears actively engaged in error monitoring.

We observed typical motor error adaptation, as indicated by the magnitude, variance, and timing of step length asymmetry during abrupt (Reisman, Block & Bastian, 2005; Vasudevan et al., 2011) and gradual adaptation (Torres-Oviedo & Bastian, 2012). Behaviorally, only a small percentage of those who performed well during abrupt adaptation also performed well during gradual adaptation. There was no correlation between abrupt and gradual adaptation performance, which means we could not predict how well a participant would adapt their gait to an abrupt perturbation schedule given step length asymmetry data from a gradual perturbation schedule (and vice versa). Participants did not fully restore gait symmetry by the end of gradual adaptation, which suggests that they did not fully compensate for the perturbation. This finding was not unexpected. Another study that used a longer 2:1 perturbation exposure (10 min) also found larger residual step length asymmetry during gradual adaptation than abrupt adaptation (Roemmich & Bastian, 2015). However, given that 2:1 perturbation in our gradual adaptation period was limited to two minutes, it is unclear if participants’ locomotor adaptation and associated electrocortical activity plateaued by the end of gradual adaptation. Notably, we observed sizable after-effects after both forms of adaptation (abrupt and gradual), which is a crucial sign of motor system involvement (Weiner, Hallett & Funkenstein, 1983).

Unlike abrupt adaptation which had a robust cortical response (Jacobsen & Ferris, 2023), there were minimal changes in gait-related spectral power during early gradual adaptation (with respect to pre-gradual adaptation). In our previous study, early adaptation to an abrupt split-belt perturbation was associated with strong alpha and beta synchronization in the sensorimotor and posterior parietal cortices, as well as increased theta power in the anterior and posterior cingulate cortices (Jacobsen & Ferris, 2023). Here, during adaptation to a gradual split-belt perturbation, there were no significant differences in event-related spectral power near the posterior cingulate, right anterior cingulate, and sensorimotor cortices, supporting our hypothesis for the right anterior cingulate cortex. The lack of spectral modulations within the sensorimotor cortex that we observed aligns with Orban de Xivry, Criscimagna-Hemminger & Shadmehr (2011) who found that the motor cortex was involved with abrupt, but not gradual, reaching adaptation. However, theta power near the left anterior cingulate fluctuated in various stages of adaptation (Fig. 8B), and alpha power near the posterior parietal cortex appeared to increase over the course of gradual adaptation (Fig. 9B). There was a significant effect of gradual adaptation subcondition on event-related spectral power for these brain regions. Therefore, we rejected our hypothesis for the left anterior cingulate cortex.

Theta oscillations in the left anterior cingulate cortex may reflect the engagement of behavioral monitoring systems. Our findings indicate that theta power near the left anterior cingulate cortex is greater during steps in the mid-ramp condition (i.e., when the belts are gradually splitting) and steps with larger kinematic errors with respect to pre-gradual adaptation. However, there were no significant differences between steps with large and small errors (i.e., large and small step length asymmetry) for any neural cluster. This suggests the left anterior cingulate can at least detect relatively large errors during gradual adaptation. Upper-limb studies have shown that midfrontal electrocortical dynamics are associated with error awareness (Ficarella, Rochet & Burle, 2019) and change as a function of error size and practice (Anguera, Seidler & Gehring, 2009). Contrarily, we did not find that theta power scales with error size, as indicated by the lack of difference between small and large errors. This may have been due to the small error range, which is much smaller during gradual adaptation than abrupt adaptation. Another relevant factor when comparing our results to previous sensorimotor adaptation studies is that walking is more cyclic and automatic when compared to discrete upper-limb movements. In the context of locomotor adaptation, theta fluctuations could reflect heightened error monitoring to maintain postural stability during challenging moments in the gait cycle, like single limb support on the faster moving belt. Prior EEG research during walking and standing has revealed theta synchronization, or a comparable decrease in event-related potentials, near the anterior cingulate cortex that may be attributed to physical loss of balance (Payne & Ting, 2020; Sipp et al., 2013), or postural threat (Payne & Ting, 2020; Peterson & Ferris, 2018; Solis-Escalante et al., 2020). Our results suggest that midfrontal theta fluctuations are not solely tied to large, high-level execution errors, as seen in many studies with abrupt perturbations, but are also relate to relatively small magnitude kinematic errors that might be below conscious perception.

While the posterior parietal plays a critical role in motor adaptation during abrupt visuomotor (Mutha, Sainburg & Haaland, 2011) and gait disturbances (Young, Parikh & Layne, 2020), its role in gradual gait adaptation is unclear. The increase in alpha power in the posterior parietal cortex during late gradual adaptation (Fig. 9B, region 3) was surprising given that the opposite effect was seen during abrupt adaptation (Jacobsen & Ferris, 2023) and the abrupt transition back to tied-belt walking (initial post-adaptation #2, Fig. 9B, region 4). Because gradual adaptation presumably mainly relies on implicit processes, we did not expect to see involvement from the posterior parietal cortex during gradual gait adaptation. Surprisingly, alpha power was greater at the end of gradual adaptation compared to pre-gradual adaptation. The changes in spectral power we observed within the posterior parietal cortex did not seem to be solely associated with adaptation because we observed increases in alpha power both as step length asymmetry increased (during gradual adaptation) and decreased (during late post-adaptation #2). While these alpha power fluctuations did not appear to be related to step length asymmetry or its variability, they could be related to another behavioral metric that we did not quantify. Alternatively, increased parietal alpha power could be related to reduced focus and attention over time, as participants had to quietly walk on the treadmill for nearly an hour. Alpha (8–13 Hz) power has been shown to systematically increase over time (minutes to hours) (Kasten, Dowsett & Herrmann, 2016), particularly near occipital and parietal regions (Benwell et al., 2019; Boksem, Meijman & Lorist, 2005), and may be associated with fatigue (Cajochen et al., 1995) (see Tran et al., 2020 for recent review). Tan, Jenkinson & Brown (2014) observed a similar effect over central brain regions while participants performed an upper-limb visuomotor adaptation task. They found alpha event-related synchronization was correlated with time spent on the task rather than error, and suggested it was more likely to be a result of systematic variation in attention or arousal as task duration increased (Macdonald, Mathan & Yeung, 2011) rather than error monitoring. In this context, we speculate that alpha power increased over the course of gradual adaptation because of reduced task engagement or fatigue, whereas the abrupt return to tied-belt walking demands a shift in attentional focus for walking stability, which would contribute to the decrease in parietal alpha power.

While it is possible that savings, or faster relearning, from the first abrupt split-belt perturbation influenced kinematic errors during gradual adaptation, we do not believe this to be the only factor influencing electrocortical activity during gradual adaptation. Roemmich & Bastian (2015) demonstrated that savings during locomotor adaptation are driven by exposure to abrupt changes in the environment and the amount of exposure to the new environment. Because abrupt adaptation always occurred first, this exposure could influence savings during the gradual adaptation period as well as the post-adaptation periods. Yet, the electrocortical responses during gradual adaptation and post-adaptation #2 were distinct, indicating that the way a perturbation is reintroduced influences brain dynamics. Spectral activity in the sensorimotor cortex was not significantly different from pre-adaptation at any stage of gradual adaptation, whereas strong alpha and beta desynchronization was present during post-adaptation #2. Near the posterior parietal cortex, alpha power gradually increased over the course of adaptation, but decreased during the beginning of the second post-adaptation period. Overall, there were more brain regions that showed significant changes in spectral power during the abrupt transition back to tied-belt walking (initial post-adaptation #2) than any stage of gradual adaptation. Differences in cortical engagement during abrupt and gradual environmental changes may explain why perturbation schedule affects transfer of walking patterns (Torres-Oviedo & Bastian, 2012).

Locomotor adaptation to abrupt and gradual perturbations may engage distinct neural processes. Many studies have investigated the neural substrates underlying implicit and explicit motor adaptation of the upper limb. Previous research has demonstrated that the neural circuits involved in learning a new movement sequence vary based on whether it is presented explicitly or implicitly (Honda et al., 1998; Pascual-Leone, Grafman & Hallett, 1994). This dichotomy might extend to locomotor adaptation involving abrupt (explicit) and gradual (implicit) adaptations, possibly engaging different distinct neural pathways. Targeting a specific neural adaptation mechanism may be of clinical benefit. For example, in poststroke patients, gradual perturbations may slightly improve the transfer of a newly learned gait pattern from a treadmill to overground compared to abrupt perturbations, possibly because small errors can be more easily credited to the person’s own movement rather than the equipment/environment (Alcântara et al., 2018). Kitatani et al. examined electromyographic coherence during abrupt and gradual leg swing perturbations, and their results suggested that behavioral responses to abrupt perturbations require more corticospinal network involvement than gradual perturbations (Kitatani et al., 2022). Our exploratory analysis revealed that, despite identical perturbations (2:1 split-belt) and similar error sizes (step length asymmetry), there was a condition effect (gradual versus abrupt) on gait-related spectral power. This effect was found in cortical clusters near the posterior parietal and posterior cingulate cortices (Fig. 12), the latter being a key region in the default mode network. Given its relationship to theta power (Prestel et al., 2018), the default mode network might be more suppressed during early abrupt adaptation than gradual adaptation. Additionally, alpha desynchronization in the posterior parietal cortex being greater during abrupt than gradual adaptation may indicate that abrupt adaptation requires more resources for sensorimotor integration and visuospatial attention (Misselhorn, Friese & Engel, 2019). There may also be other biomechanical differences (e.g., stability, muscle activity) that are related to these electrocortical differences between abrupt and gradual adaptation that we did not explore. Given the relationship between frontal theta and balance ability (Payne & Ting, 2020; Sipp et al., 2013; Solis-Escalante et al., 2020; Stokkermans et al., 2022), it would be beneficial for future work to compare anterior cingulate theta with indices of gait stability. Because the condition order was not randomized, we cannot draw firm conclusions on comparisons between abrupt and gradual gait perturbations. We suggest that future studies investigate differences in electrocortical activity between abrupt and gradual gait perturbation schedules by randomizing the condition order or using two separate groups.

There are a few important limitations to address. Overall, our results suggest that increased gait perturbation and step length asymmetry magnitudes were not by themselves sufficient to produce measurable changes in electrocortical activity near the sensorimotor, posterior cingulate, and right anterior cingulate cortices. However, we cannot conclude that these brain regions are completely uninvolved in gradual adaptation, as the lack of detectable changes may reflect a sensitivity limit of EEG. Additionally, results comparing abrupt and gradual adaptation should be interpreted with some caution because the condition order was not randomized, as gradual adaptation was not the primary focus of our original study. There may have been behavioral savings from abrupt adaptation that transferred and affected the rate of gradual adaptation, but we expect this to be minimal because participants only experienced one split-belt period prior to gradual adaptation and still exhibited errors. Nonetheless, these results indicate there could be distinct cortical representations for different perturbation schedules.

Adaptation can involve a complex interplay of various learning mechanisms, such as explicit and reinforcement learning, making it challenging to fully isolate implicit learning. Despite our efforts to minimize participants’ awareness of the gradual gait perturbation, it is still possible that other small sensory cues conveyed information about belt symmetry. We discouraged participants from looking downwards at the treadmill but could not completely prevent them from doing so. If a participant saw the differing belt speeds or accidentally drifted mediolaterally, they may have realized the belt speeds were different. This could have triggered explicit adaptation mechanisms due to visual feedback of belt speeds or the mechanical perturbation caused by the foot being placed on both belts simultaneously. Although our results from the gradual adaptation condition cannot be fully attributed to implicit learning, we do not think that explicit learning was overall a large driver of adaptation. This is because gradual adaptation did not have the same large switching cue to trigger participants to immediately recall extrinsic information from abrupt adaptation. Future studies could use glasses that block vision of the treadmill and a divider between belts to reduce contextual clues during gradual adaptation, although it is unclear how introducing these devices would affect gait and electrocortical activity.

There are other supraspinal mechanisms that may be involved in gradual adaptation that we were unable to measure with our methodology. High-density EEG in mobile subjects is not likely to record cerebellum, basal ganglia, or brain stem electrical activity. There are some studies that suggest some of these may be recorded in sitting, unmoving subjects (Fahimi Hnazaee et al., 2020; Samuelsson et al., 2020), but quantifiable activity from those areas during walking has not been validated. Future locomotor adaptation studies could use mobile EEG to explore the questions regarding difference perturbation schedules and the relationship between the cerebellum and cerebral cortex. This could be achieved by studying clinical populations with cerebellar damage (e.g., cerebellar ataxia) or participants undergoing electrical cerebellar stimulation while abruptly or gradually introducing gait perturbations.

Conclusions

We quantified behavioral and electrocortical responses to a gradually introduced gait perturbation during treadmill walking to gain insight on the electrocortical dynamics of locomotor adaptation. We hypothesized there would be no change in gait-related theta power (4–7 Hz) in the anterior cingulate cortex across stages of gradual gait adaptation relative to pre-adaptation because a gradually changing belt speed ratio would not challenge gait stability and evoke large stepping errors. We found that a gradually introduced split-belt perturbation does not elicit large kinematic errors, as measured by step asymmetry, or widespread changes in electrocortical dynamics (<30% of cortical clusters changed) in the same way as abrupt transitions. Contrary to our hypothesis, there were significant increases in theta spectral power in the left anterior cingulate that suggest its involvement in error monitoring during gradual gait adaptation. However, we did not observe any significant differences between relatively large and small stepping errors, despite evidence from previous literature suggesting theta increases with the perturbation intensity and its associated destabilizing effects (Aiden et al., 2019; Dietz, Quintern & Berger, 1984; Goel et al., 2018; Mochizuki et al., 2010; Solis-Escalante et al., 2020). Our findings provide evidence that small gait perturbations and errors are reflected in neural oscillations during human walking. The findings provide a good benchmark for identifying dysfunctional brain dynamics in patient populations that have difficulty adapting their gait to similar perturbations.

The authors thank Nicole Esposito for helping with data collections. The authors also thank JiHo Han, Rohan Lingam, Ami Patel, and John Prieschl for assistance with processing the motion capture data.

Additional Information and Declarations

Competing Interests

Author Contributions

Ethics

Data Availability

The authors declare there are no competing interests.

Noelle A. Jacobsen conceived and designed the experiments, performed the experiments, analyzed the data, prepared figures and/or tables, authored or reviewed drafts of the article, and approved the final draft.

Daniel Perry Ferris conceived and designed the experiments, authored or reviewed drafts of the article, and approved the final draft.

The following information was supplied relating to ethical approvals (i.e., approving body and any reference numbers):

The University of Florida granted Ethical approval to carry out the study within its facilities

The following information was supplied regarding data availability:

The data are available at OpenNeuro: Noelle A. Jacobsen and Daniel P. Ferris (2023). Mobile EEG split-belt walking study. OpenNeuro. [Dataset] doi: 10.18112/openneuro.ds004475.v1.0.3.

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
