# Peer review of "Electrocortical theta activity may reflect sensory prediction errors during adaptation to a gradual gait perturbation"

_PeerJ, doi:10.7717/peerj.17451_

## Round 0.1 · original submission · Major Revisions

The reviewers requested improvements of clarity in study design and reporting. Please, address these concerns.

·

Basic reporting

Line 167 - 'Figure' not 'Figures'
Line 171 - Delete the sentence starting with "These data were synchronized..."
Line 197 - Delete 'followed'
Line 243 - Delete the portion of the sentence regarding showing the participant the raw EEG data stream. Include just the portion that they were instructed to refrain from jaw clenching, excessive eye blinking, and neck muscle tension.
Line 255 - In Figure 2-B, align the y-axis with the y-axis in Figure 2-A.
Line 265 - Format this equation the same way you formatted equation (1) below.
Line 268 - Delete sentence starting with "We used the normalized step length...", this is assumed.
Line 373 - 'Adaptation' not 'adaption'
Line 400 - 'asymmetric' not 'asymmetry'
Line 418 - 'Pre-gradual adaptation' not 'gradual adaptation'
Line 507 - I do not see a region labelled 1 in Figure 9-B.
Line 568 - I do not see a region 1 in Figure 13-B.
Line 570 - I do not see region 2 in Figure 13-B.
Line 576 - I do not see region 3 in Figure 13-B.

Experimental design

Some descriptions in the methods and results are unclear, refer to these lines:
Line 176 - Do you include any measures to ensure they were not looking down at the belts? For instance, visual field-limiting goggles or a physical barrier?
Line 180 - How were these values determined? What is your rationale for not having speeds based on the participant’s preferred speed?
Line 192 - Figure 2 would be more clear if the pre-adaptation period showed the tied-belt condition at the 3 different speeds.
Line 197 - How did you determine what was an appropriate amount of time for the aftereffects to wash out?
Line 204 - This somewhat goes against the rationale for your hypothesis. If I am understanding correctly, you expect there to be small stepping kinematic errors during the gradual perturbation, so you should not expect to see adaptation or corresponding increases in mid frontal theta oscillations. However, in this sentence, you state that you do expect to see adaptation in the gradual perturbation.
Line 233 - What were the hardware and software issues? Could these issues compromise the accuracy of recording the electrode locations?
Line 241 - What is PowerFlex? I am assuming this is a brand name for some kind of adhesive. In this case, it would be best to use generic terms.
Line 256 - Do you only look at these sub conditions for the post-gradual adaptation period? If you also look at them for the post-abrupt adaptation, please highlight these regions in the figure and include it in the text.
Line 260 - This description is unclear, perhaps a figure would make it more clear.
Line 278 - This contradicts a sentence you said previously: “This was followed by a 10-minute post-adaptation period (post-adaptation #1) to ensure after-effects were washed out”. Did the aftereffects wash out or not? Additionally, it would be interesting to see if the lack of washed out aftereffects you see in the pre-gradual adaptation period are actually a natural asymmetry for the participant. You could compare the step length asymmetry between the pre-abrupt adaptation and pre-gradual adaptation periods to see if they are both non-zero (and not significantly different). If so, then it could be a natural asymmetry that the participant has and not a result of the imposed adaptation.
Line 280 - It is unclear why you need to calculate the belt speeds, are they not set?
Line 367 - In Figure 4-C, both signals abruptly go towards 0.9 m/s shortly after stride #2150. That portion should align with the late 2:1 split condition in Figure 4-A (the last 30 steps where the belts are at 1.2 m/s and 0.6 m/s). Could this convergence back to 0.9 m/s be you actually looking at the initial steps of the post-adaptation #2 condition?
Line 488 - Unclear what a semi-transparent mask means
Line 560 - Be more specific in regards to which phase of the gait cycle and within what frequencies.

Validity of the findings

Line 443 - I am not fully convinced. Looking at Figure 2-B and comparing the step length asymmetry between the pre-gradual adaptation period and the gradual adaptation period, I do not necessarily see a difference in magnitude. The gradual adaptation period does not have an initial rapid change like the abrupt adaptation or the post-adaptation periods.
Line 452 - A potentially more effective analysis to see if the participants who were good at adapting to the abrupt perturbation were also good at adapting to the gradual perturbation could be to plot the gradual performance score vs the abrupt performance score and look at the correlation between the two.
Line 494 - How to interpret Figure 8-B is not immediately clear. To start, the ranges for the color gradient that represents the difference in power are not consistent across clusters. For example, the range in difference in power for the left anterior cingulate cluster is [-1, 1] with very dark blue representing -1 and very dark red representing 1. Now, when you go to look at the right anterior cingulate, the range is [-0.8, 0.8] where very dark blue now represents -0.8 and very dark red now represents 0.8. Second, the parts of the figure that show significant differences are not immediately clear. The part circled and labelled 1 helped, but the parts you reference in the right posterior cingulate row are not labelled and are hard to distinguish. Third, I have a hard time being convinced that the three parts of the figure you pointed out are the only parts that show significant differences. For example, in the left anterior cingulate row and the late 2:1 column, I see areas of red that are close in color to those in the mid ramp column that you point out (accounting for the semi-transparent mask), but the area in the late 2:1 column does not have significant differences while the area in the mid ramp column does.
Line 537 - If there is no significance, then you cannot make this claim.
Line 622 - I am not following here. How can you have peak theta power during larger kinematic errors but have no significant differences between large and small errors.
Line 623 - It is unclear how you arrive at this conclusion based on the previous sentence.
Line 649 - If it is not significantly different, then you cannot say it is greater.

Reviewer 2 ·

Basic reporting

Figure 5 legend– Consider altering the description of “one being good and zero being bad” to a more descriptive meaning of the values, e.g. more and less symmetrical. It also appears that the abrupt adaptation score is used to rank the participants, but is not in the description.
Figure 6 legend – Please clarify the source of the template brain MRI. For example, is this an example participant or from open source?
Figure 7 and methods – Can the authors identify the range and the mean amount of variance explained at each region?
Can the authors provide their interpretation for the scenarios of an absent dipole for any region of interest within a specific individual? Is this due to technical limitations or neurophysiologic reasons? This will help inform future studies that may employ similar methods in patient populations. Here, it appears that not having an identifiable response in a region of interest can be considered neurotypical.
Figures 8 and 9 and 11 – The semitransparent mask is a good thought, but difficult to read. Can the authors increase the transparency to increase the contrast? Additionally, the left ACC appears to be the only circled region signifying discussion in the text. However, there appear to be other significantly different regions from the pre-gradual adaptation that were mentioned in the introduction, such as the PCC. The authors may consider circling significant regions in the figure and assigning them serial numbers for discussion in the text.
Figure 10 – Clarify how the y axes is normalized.
The two distinct aims that attempt to delineate initial errors and the magnitude of errors are likely capturing the same mechanisms in the current experimental design. It is unclear how differences in findings between these two analyses should be interpreted.
Figure 13 – The circled regions statement here and elsewhere does not appear relevant and is confusing to the reader. Additionally, it implies that absences of differences between conditions were not discussed in the text.

Experimental design

A key limitation is the non randomized order of the abrupt and gradual adaptation conditions. However, I believe the authors have done a nice job of identifying this limitation and bringing this point to the forefront to inform reader interpretation of the findings presented.
A key strength to the experiment is the use of the dual-electrode system.

Validity of the findings

• Much of the motivation for the present analyses involves the roll of the ACC in error detection. The authors’ rationale and interpretations of the findings would be strengthened if they included a “control” brain region that would not be changed during treadmill adaptation, e.g. occipital.
• Additionally, it is interesting that the authors do not identify SMA as a region of interest, given the EEG response following balance perturbations has been localized to SMA (see Marlin et al 2014). Did the authors locate dipoles within SMA and are they able to characterize these SMA response profiles in the context of their current analyses?
• The authors briefly mention the point that step length symmetry may not be the variable that individuals (some or all) are attempting to optimize during adaptation (lines 725-727). I thank the authors for making this point. Given the consistent link between EEG activity and balance/walking ability shown in previous literature (in young neurotypical and patient populations), it would be informative to test whether the magnitude of the initial change in stability within each condition was associated with the magnitude of ACC change.

Additional comments

Lines 641-665 – Regarding the discussion point on the unexpected posterior parietal cortex findings – it is unclear why this would be the brain region implicated in changes in attention over time. Can the authors please expand on this explanation for PPC?

Lines 719 – 720 – Can the authors reference the figure/results to which they are specifically referring to?

Please change the term “subject” to participant throughout.

Line 156 – Please change term “healthy young adults” to “young, neurotypical” or something along those lines.
Line 702 – same comment as above.

---

## Round 0.2 · accepted · Accept

Thank you for addressing the comments.

·

Basic reporting

no comment

Experimental design

no comment

Validity of the findings

no comment

Reviewer 2 ·

Basic reporting

Revision seems ok.

Experimental design

Revision seems ok.

Validity of the findings

Revision seems ok.

Additional comments

Revision seems ok.